# Role of POT1 in Human Cancer

**DOI:** 10.3390/cancers12102739

**Published:** 2020-09-24

**Authors:** Yangxiu Wu, Rebecca C. Poulos, Roger R. Reddel

**Affiliations:** 1Cancer Research Unit, Children’s Medical Research Institute, Faculty of Medicine and Health, The University of Sydney, Westmead NSW 2145, Australia; ywu@cmri.org.au; 2ProCan® Cancer Data Science Group, Children’s Medical Research Institute, Faculty of Medicine and Health, The University of Sydney, Westmead NSW 2145, Australia; rpoulos@cmri.org.au

**Keywords:** POT1, telomere, telomere length, cancer, shelterin, telomerase, mutation, genomic instability, alternative lengthening of telomeres

## Abstract

**Simple Summary:**

The segmentation of eukaryotic genomes into discrete linear chromosomes requires processes to solve several major biological problems, including prevention of the chromosome ends being recognized as DNA breaks and compensation for the shortening that occurs when linear DNA is replicated. A specialized set of six proteins, collectively referred to as shelterin, is involved in both of these processes, and mutations in several of these are now known to be involved in cancer. Here, we focus on Protection of Telomeres 1 (POT1), the shelterin protein that appears to be most commonly involved in cancer, and consider the clinical significance of findings about its biological functions and the prevalence of inherited and acquired mutations in the *POT1* gene.

**Abstract:**

Telomere abnormalities facilitate cancer development by contributing to genomic instability and cellular immortalization. The Protection of Telomeres 1 (POT1) protein is an essential subunit of the shelterin telomere binding complex. It directly binds to single-stranded telomeric DNA, protecting chromosomal ends from an inappropriate DNA damage response, and plays a role in telomere length regulation. Alterations of *POT1* have been detected in a range of cancers. Here, we review the biological functions of POT1, the prevalence of *POT1* germline and somatic mutations across cancer predisposition syndromes and tumor types, and the dysregulation of POT1 expression in cancers. We propose a framework for understanding how POT1 abnormalities may contribute to oncogenesis in different cell types. Finally, we summarize the clinical implications of POT1 alterations in the germline and in cancer, and possible approaches for the development of targeted cancer therapies.

## 1. Introduction

Telomeres are specialized nucleoprotein structures that cap the ends of linear chromosomes, consisting of a complex of DNA, RNA and proteins. In mammals, telomeric DNA contains tandem repeats of the hexameric DNA sequence 5′-TTAGGG-3′ [1], which is mostly double-stranded (ds), but ends with a single-stranded (ss) overhang of the G-rich strand, ranging from 50 to 400 nucleotides (nt) in length. Telomeres play an important role in maintaining genome stability [2,3]. Moreover, telomeres control cellular replicative potential, because, in most somatic cells of multicellular eukaryotes, telomeric DNA gradually shortens with each cell cycle, due, in part, to the end-replication problem, whereby the linear ends of chromosomes are incompletely replicated [4,5] and, in part, due to the enzymatic processing of the C-rich strand [6,7]. Replicative senescence [8] ensues when telomeres shorten below a certain length threshold [9,10].

Most cancer cells have an activated telomere maintenance mechanism (TMM) which prevents the excessive shortening of telomeres [11]. Two TMMs have been identified to date, namely telomerase and alternative lengthening of telomeres (ALT) [12,13]. Telomerase is a ribonucleoprotein reverse transcriptase that synthesizes new telomeric DNA on a template within its RNA subunit [14,15]. ALT is a homologous recombination (HR)-mediated DNA repair mechanism that synthesizes new telomeric DNA using existing telomeric DNA as a template [16]. The upregulation of a TMM is a core event in the acquisition of unlimited replicative potential (i.e., cellular immortality), one of the hallmarks of cancer [17].

Since telomeres are associated both with genome stability and cellular immortalization, the study of telomere biology is important for understanding cancer risk, diagnosis, prognosis and the outcome of therapy, and for devising new therapies. In this review, we focus on the role of one of the telomere-associated proteins, Protection of Telomeres 1 (POT1), in telomere function and cancer. Germline and somatic POT1 mutations and the dysregulation of POT1 expression have been detected across different cancer types, most prevalently in cutaneous melanoma and squamous cell carcinoma, angiosarcoma, non-small-cell carcinoma of the lung and chronic lymphocytic leukemia (CLL). We therefore summarize what is known about the normal function of the POT1 protein, and the nature of the POT1 alterations in human cancer, and consider the insights this provides into cancer biology and potential clinical implications.

## 2. Functions of POT1

### 2.1. Interactions of the Shelterin Complex with Telomeric DNA

Telomeric DNA is bound by a protein complex called shelterin, containing six subunits—the TRF1, TRF2, RAP1, TIN2, TPP1 and POT1 proteins—encoded by the Telomeric Repeat Binding Factor 1 *(TERF1)*, Telomeric Repeat Binding Factor 2 *(TERF2)*, Telomeric Repeat Binding Factor 2 Interacting Protein *(TERF2IP)*, TERF1-Interacting Nuclear Factor 2 *(TINF2)*, Adrenocortical Dysplasia Homolog *(ACD)* and *POT1* genes, respectively (reviewed by [18]) (Figure 1). TIN2 is a central factor in the shelterin structure, linking TRF1, TRF2 and TPP1. TRF1 and TRF2 bind directly to ds telomeric DNA. RAP1 interacts with TRF2, and POT1 interacts with both TPP1 and ss telomeric DNA. Shelterin plays a vital role in telomere function by remodeling telomeric DNA into a protected structure and managing the regulation of telomere length (reviewed by [19]). Loss or malfunction of shelterin proteins results in uncapped telomeres, which induce genome instability and, ultimately, cellular aging or apoptosis (reviewed by [20]). 

Protection of Telomeres 1 (POT1) is an essential subunit in the shelterin complex [19,21,22]. In humans, POT1 is a 634 amino acid protein, encoded by a gene on chromosome 7q31.33. Human POT1 recognizes and binds telomeric ss DNA via both of the oligonucleotide/oligosaccharide-binding (OB) fold domains (OB1 and OB2) in its N-terminus [23,24], and it is the only shelterin protein with this function. The localization of POT1 to telomeres requires its interaction with TPP1 through its C-terminal half, which contains a third OB fold (OB3) that is split by a Holliday junction resolvase-like domain [25,26,27,28] (Figure 1A). Of the five isoforms of POT1 generated by alternative splicing, only the full-length isoform V1 is able to bind to TPP1; there is little information available about V4, whereas V2, V3 and V5 have been shown to associate with telomeres and regulate overhang length [29] (Figure 1B). TPP1 and POT1 are ten-fold less abundant than the other shelterin proteins [30], so it is assumed that some shelterin complexes lack these two proteins, but there is also a ten-fold excess of TPP1/POT1 compared to its telomeric ss DNA binding sites, suggesting that these sites are mostly occupied. One POT1 molecule occupies two telomere repeat sequences, i.e., 12 nt [31].

The ends of telomeres fold back and form a loop structure, called a t-loop [32], presumably due to the annealing of the terminal ss DNA with the complementary strand of telomeric ds DNA. POT1 does not appear to be required for t-loop formation [33]. However, one of POT1′s functions is to control the sequence at the junction of the telomeric ds DNA with the ss DNA overhang [21] (Figure 1A).

In contrast to the single POT1 protein in humans, mouse shelterin contains two POT1 proteins, mPOT1a and mPOT1b, that are closely related to each other, resulting from a gene duplication event [34,35]; study of these paralogs has facilitated dissection of POT1 function. mPOT1a represses the Ataxia Telangiectasia- and Rad3-related (ATR)-mediated DNA damage response machinery at telomere regions, contributing to the prevention of telomere termini being recognized as DNA breaks. mPOT1b regulates the length of the single-stranded telomeric 3′ G-overhang by controlling 5′ nucleolytic processing of the C-rich strand. Human POT1 has been shown to possess both of these functions.

### 2.2. Repression of the ATR-Mediated DNA Damage Response

POT1 binds ss telomeric DNA, while avoiding binding to telomeric transcript Telomeric Repeat-containing RNA (TERRA) [38]. When bound to the 3′telomeric overhang, POT1 is able to inhibit inappropriate DDR signaling. The proposed mechanism is such that POT1 is able to block the much more abundant Replication Protein A (RPA) from coating telomeric ss DNA, thereby inhibiting the recruitment of ATR and the consequent DNA damage response (DDR) [22,39,40,41]. It has been proposed that Heterogeneous Nuclear Ribonucleoprotein A1 (hnRNPA1) protein removes bound RPA, an abundant heterotrimeric ss DNA-binding protein, in a process involving TERRA, and is replaced by POT1 [40]. Inhibition of ATR-mediated DDR is dependent on POT1’s ability to bind TPP1 and on TPP1′s ability to interact with TIN2 [42,43], suggesting that it is the tethering of POT1 to the shelterin complex that allows it to outcompete RPA for binding at the telomere even though RPA is much more abundant and has a similar affinity for telomeric ss DNA. Moreover, POT1 binds to sites adjacent to G4 structures, which may also increase its ability to compete with RPA at telomeres [44].

### 2.3. Control of 5′ Nucleolytic Processing of the C-Rich Strand: Interactions with CST

The replication product of leading-strand telomeric DNA synthesis may often be a blunt end, so the generation of a ss DNA 3′ overhang requires nucleolytic processing of the 5′ strand. The first indication of POT1′s involvement in controlling this processing was the observation that it controls the specificity of the sequence at the end of the 5′ strand [21]. The mechanism of 5′ strand processing is best understood for mouse telomeres, where an intricate series of steps controlled by TRF2 and POT1b has been elucidated, involving nucleolytic attack by the Apollo and Exonuclease 1 proteins, and fill-in DNA synthesis mediated by polymerase (Pol)α/primase and its accessory factor, the Ctc1/Stn1/Ten1 (CST) complex (reviewed in [45]). In human cells, the CST complex binds both POT1 and TPP1, whereas it binds POT1b alone in the mouse, and studies of a POT1 mutation in two siblings with Coats Plus (CP) syndrome, one of the inherited telomere biology disorders (TBD), indicate that POT1 may promote CST-mediated fill-in [46].

### 2.4. Control of Telomerase-Dependent Telomere Lengthening

POT1 controls telomerase-dependent telomere elongation. This is achieved in part by the TPP1–POT1 heterodimer recruiting telomerase to telomeres and enhancing its processivity [24,47,48,49,50,51]. However, POT1 may negatively modulate telomere length by competing with telomerase for binding to the 3′ end of telomeric ss DNA, and sequestering the last guanine base into a hydrophobic pocket that makes it inaccessible to telomerase [24,37,52]. Hence, inhibition of POT1, or the presence of POT1 mutants that lack DNA binding but retain TPP1 binding and thus have a dominant-negative function, may lead to extensive upregulation of telomere length in cells that have telomerase activity [24,28,47,48,49,50,51].

The TPP1–POT1 heterodimer may also control telomerase-dependent telomere lengthening through recruitment of CST, which is also a negative regulator of telomerase: depletion of CST proteins or overexpression of a dominant-negative mutant of CTC1 may result in telomere lengthening in some telomerase-positive human cell lines [40,53,54,55]. The mechanism of this inhibition could potentially be due, in part, to the ability of CST to bind to telomeric DNA, like POT1, and thereby block access to telomerase [54]. However, the fill-in reaction by Polα/primase, which is facilitated by CST, may also inhibit telomerase activity. The evidence for this includes the observation that telomeres become lengthened when polα is partially inhibited by aphidicolin [56]. The CP POT1 mutant referred to above (an S322L substitution) was able to bind DNA and TPP1, and block ATR signaling, but was defective in inhibiting telomerase and therefore resulted in extended 3′ telomeric overhangs, defective C-strand maintenance, and stochastic telomere truncation events that could be healed in telomerase-positive cells. These observations are consistent with the hypothesis that POT1 S322L is a separation-of-function mutant, and that POT1 regulates telomerase, at least, in part, through the recruitment of CST to telomeres [46] (27).

### 2.5. Unfolding of G-Quadruplexes

Telomeric DNA is G-rich, and it can therefore form G-quadruplex (G4) structures by Hoogsteen base pairing, and the long ss DNA 3′ overhang can form multiple G4 units. Biophysical studies of ss telomeric DNA up to 196 nt in length have shown that it folds into structures with the maximum possible number of G4 units and with the longest ss gaps of 3 nt [57]. G4 DNA obstructs DNA replication, but several studies have shown that POT1 unfolds G4 DNA to produce a ss DNA-POT1 complex [58,59,60,61,62]. A recent study showed that POT1 can unfold any conformational form of telomeric G4, with a rate of unfolding that is similar to the intrinsic unfolding rate for each of these forms, suggesting that POT1 traps telomeric DNA in its ss form [57].

### 2.6. Repression of Homologous Recombination at Telomeres

The long tracts of tandem hexameric repeats at telomeres make sister telomeres very vulnerable to homologous recombination (HR), with unequal exchanges having the deleterious effect of shortening the replicative potential of one of the daughter cells. The repression of telomeric HR is therefore presumably very important, and in mouse cells, POT1a, POT1b, and RAP1 play a role in achieving this by mechanisms which are not well understood [63,64]. How telomeric HR is repressed in human cells and, conversely, how telomeric HR is upregulated in cancer cells that use the ALT TMM despite the presence of shelterin proteins, are understood to an even lesser extent.

### 2.7. Summary of POT1 Functions

POT1 is an integral member of the shelterin complex, and its specialized functions primarily involve the generation, stabilization and protection of telomeric ss DNA. Its role in the process of generating the 3′ G-rich ss overhang is mediated through its interaction with the Polα/primase accessory factor—the CST complex. POT1 controls the length of the 3′ ss overhang in part through the Polα/primase fill-in reaction, and in part through limiting access of telomerase to the telomere by sequestering the terminus of the overhang and by recruiting CST which also competes with telomerase for access to the telomere. It stabilizes and protects telomeric ss DNA by binding to it directly (one POT1 molecule every 12 nt), by unfolding the G4 structures which telomeric ss DNA is highly prone to form, and possibly by contributing to the repression of HR-mediated repair reactions. POT1 suppresses the ATR-mediated DNA response indirectly, by its role in overhang production, which allows the formation of t-loops and, more directly, by blocking RPA from coating telomeric ss DNA, thereby inhibiting the recruitment of ATR. 

Although there is more to discover about the mechanisms of POT1’s functions, it is already abundantly clear that POT1 has a pivotal role in the control of telomere length and stability. It may therefore be predicted that malfunction of POT1 could potentially be involved in the dysregulation of telomere length and in genomic instability, both of which are features of cancer. Moreover, POT1’s role in modulating telomerase activity suggests that malfunction of POT1 would affect cells that express telomerase differently from those that are telomerase-negative.

## 3. Germline and Somatic *POT1* Mutations in Cancer

*POT1* germline mutations have been identified across various cancer types, most notably in melanoma, chronic lymphocytic leukemia, angiosarcoma and glioma. *POT1* is also somatically mutated in a number of cancer types. A recent pan-cancer study of POT1 mutations categorized genetic variants according to the standards of the American College of Medical Genetics and Genomics, and referred collectively to all POT1 variants that are neither “presumed benign” nor “benign” as “non-benign”. By this classification, 2.9% of tumors (*n* = 1834/62,368) were found to have non-benign *POT1* mutations [65]. Among the tumor categories with more than 50 cases, angiosarcoma (23.3%, *n* = 20/86) and cutaneous squamous cell carcinoma (9.1%, *n* = 20/220) had the highest prevalence of *POT1* mutations [65] (Figure 2).

### 3.1. Melanoma

Cutaneous melanoma commonly arises as a result of exposure to ultraviolet light. Around 10% of people with melanoma have a family history of the disease, and *CDKN2A* is the most highly penetrant susceptibility gene in familial melanoma. Mutations in *CDKN2A* are detected in 10–40% of familial melanoma cases, with *CDK4* also an established melanoma predisposition gene (reviewed by [66,67]). *POT1* has been reported as among the top five predisposition genes in familial melanoma (reviewed by [68]). In a study of *CDKN2A/CDK4* wild-type melanoma-prone families recruited from Australia, UK and the Netherlands, four pedigrees were found (3.8%, *n* = 4/105) in which melanoma co-segregated with *POT1* germline variants (p.Tyr89Cys, p.Gln94Glu, p.Arg273Leu, and c.1687-1G > A) [69]. 

In a similar Italian cohort [70], seven pedigrees (12.5%, *n* = 7/56) were identified with three distinct *POT1* mutations (p.Ser270Asn, p.Arg137His and p.Gln623His). The p.Ser270Asn mutation was most common, having been detected in five unrelated families. In the same study, two other rare recurrent *POT1* variants (p.Asp224Asn and p.Ala532Pro) were identified in pedigrees from the USA and France (0.6%, *n* = 2/305) [70]. In a Spanish study of melanoma-prone families without CDKN2A or *CDK4* mutations, four pedigrees (1.7%, *n* = 4/228) were identified with further distinct *POT1* variants (p.Ile78Thr, p.Glu344*, c.255G > A and p.Asp598Serfs*22) [71]. The *POT1* p.Ile78Thr variant was also discovered in three Jewish families [72]. Exonic *POT1* mutations were also found in Austrian familial melanoma cases (1.5%, *n* = 2/133) [73], but not in an Italian cohort of familial and sporadic multiple primary melanoma (MPM) cases [74]. Furthermore, in Dutch melanoma-prone pedigrees (*n* = 451), two missense variants in the shelterin subunits *ACD* and *TERF2IP* were identified, but none in the *POT1* gene [75]. These results demonstrate that the prevalence of *POT1* germline mutations is variable across cohorts, but *POT1* nevertheless appears to be one of the most frequently mutated genes arising in melanoma families. Somatic mutation data obtained from The Cancer Genome Atlas (TCGA) Pan-Cancer cohort through CBioPortal [76,77] suggests that *POT1* is mutated in approximately 4% of sporadic melanomas (*n* = 444). This figure comprises both missense and truncating mutations (2.3%, *n* = 10/444), as well as gene copy number amplifications (1.8%, *n* = 8/444). Similarly, Shen et al. found that non-benign *POT1* variants were observed in 3.7% of melanoma cases examined (*n* = 64/1735) [65] (Figure 2). 

### 3.2. Chronic Lymphocytic Leukemia (CLL)

CLL is the most common leukemia arising in Western countries, and those with a family history have a higher chance of developing the disease [78,79]. Both germline and somatic *POT1* mutations have been identified in CLL patients. In a study of pedigrees with a family history of CLL, four *POT1* germline mutations (p.Tyr36Cys, p.Gln376Arg, p.Gln358SerfsTer13 and c.1164-1G > A) were found to co-segregate with CLL occurrence (6%, *n* = 4/66). Two variants in shelterin genes, TERF2IP and ACD, were also identified. The onset of CLL in individuals with germline mutations in shelterin subunits occurred at an earlier age than in the population overall, suggesting that shelterin mutations do contribute to leukemia susceptibility [80]. 

Recurrent somatic mutations in *POT1* have also been identified in CLL cases, with between 3 and 7% of patients carrying *POT1* variants according to whole-exome and whole-genome sequencing data [81,82,83,84]. Using targeted sequencing, mutant *POT1* has also been identified in roughly 2.5–10% of CLL cases [85,86,87,88]. Ramsay et al. [89] detected *POT1* somatic mutations in approximately 3.5% of patients (*n* = 12/341), with the frequency of *POT1* mutations increasing to as high as 9% in a subgroup of patients with wild-type immunoglobulin heavy chain variable (*n* = 9/98) [89]. *POT1* mutations have also been noted in relapsed CLL patients, at much higher frequencies than in primary tumors (13.1%, *n* = 8/61) [90]. Interestingly, *POT1* mutations have been associated with CLL formation after radiation exposure, as *POT1* was the most frequently mutated gene in Ukrainian Chernobyl cleanup workers who developed CLL (25%, *n* = 4/16) [91]. 

### 3.3. Angiosarcoma

Angiosarcoma is a rare cancer, accounting for approximately 2% of all soft tissue sarcomas (reviewed by [92]). In a study of Spanish *TP53*-negative Li–Fraumeni-like (LFL) families, the p.Arg117Cys *POT1* germline mutation was associated with familial cardiac angiosarcoma (CAS) [93]. *POT1* variants were subsequently observed in 10% of sporadic CAS (*n* = 1/10), and 10% of sporadic cardiac sarcoma (*n* = 2/20), while 20% of LFL families with members who suffered from angiosarcoma had a *POT1* germline mutation (*n* = 2/10) [94]. Another somatic variant of *POT1* (p.Arg432*) has been identified in a small CAS cohort (20%, *n* = 1/5) [95]. Shen et al. [65] found that angiosarcoma was 11 times more likely to harbor mutated *POT1* than other cancers overall. 

### 3.4. Glioma

Germline *POT1* mutations have been associated with glioma, a malignant brain tumor of glial tissue. For example, two *POT1* germline mutations (p.Gly95Cys and p.Glu450*) were identified among one cohort of familial glioma cases (3.6%, *n* = 2/55), and an additional germline variant (p.Asp617Glufs) was discovered in another cohort (0.4%, *n* = 1/246) [96]. In a subsequent study, the germline and somatic molecular profiles of cancer samples were generated from 20 unrelated familial glioma patients, and these same *POT1* inherited variants were identified in two cases (10%, *n* = 2/20), in addition to an acquired *POT1* mutation at p.Arg117His (5%, *n* = 1/20) [97]. 

### 3.5. Other Cancers

Germline mutations of *POT1* have also been identified in other familial cancers. An analysis of thousands of familial colorectal cancer cases alongside healthy controls identified three *POT1* germline mutations (p.Asp617GlufsTer9, p.Arg363Ter and p.Asn75LysfsTer16) [98]. *POT1* was designated as one of three novel candidate colorectal cancer susceptibility genes, alongside *MRE11* and *POLE2* [98]. In Hodgkin’s lymphoma families, two *POT1* germline variants (p.Asp224Asn and p.Tyr36His) were identified (4.8%, *n* = 2/41), with p.Asp224Asn already observed in familial melanoma [70,99]. Another inherited mutation was observed in people with a family history of cancer who developed malignant mesothelioma (0.2%, *n* = 1/432) [100]. Four *POT1* mutations were identified significantly more frequently in breast cancer patients (*n* = 1067) than in healthy controls (*n* = 1110) [101]. Moreover, in a recent study focusing on pathogenic osteosarcoma germline mutations, five *POT1* germline variants (p.Asp42Tyr, p.Gln376Arg, p.Leu69Phe, p.Asp617fs; p. and c.670G > A) were detected (*n* = 5/1004) [102]. In those with European ancestry, *POT1* germline mutation frequency was statistically significantly enriched in those with disease (0.5%, *n* = 4/732), indicating a potential association between *POT1* germline variants and risk of developing osteosarcoma [102]. Furthermore, one case–control study has shown that the presence of the rs10244817 variant in *POT1* is significantly associated with lung cancer [103], and a single nucleotide polymorphism (SNP) near *POT1* (rs116895242) is associated with a reduced likelihood of acquiring colorectal, ovarian, and lung cancer [104]. Finally, a highly penetrant *POT1* mutation (p.Lys90Glu) was recently discovered in a large US family, where carriers suffered from a range of different cancer types including multiple primary melanoma, thyroid cancer, breast cancer and others [105]. This finding suggests a potentially broad role for *POT1* germline mutations in conferring cancer susceptibility across cancer types. This germline variant was recently reported as a somatic mutation in CLL [89]. 

Regarding somatic mutations, several *POT1* variants have been detected in a mantle cell lymphoma cohort (5.4%, *n* = 3/56) [106], and somatic *POT1* mutations might also contribute to the formation of sporadic parathyroid tumors [107,108]. Mutant *POT1* has been found with a relatively high prevalence in triple-negative breast cancer (6.7%, *n* = 4/59) [26]. The prevalence of germline and sporadic POT1 mutations in various tumor types is summarized in Table 1.

### 3.6. POT1 Mutations and Mechanisms of Oncogenesis

As described above, both germline and somatic mutations have been associated with a range of tumor types. Interestingly, many of the *POT1* variants reported in cancer are in the N-terminal OB-fold domains, which constitute the telomeric DNA binding area [69,70,80,88,93,96,98,99,105]. Functional studies suggest that these disruptive OB1/OB2 mutations may act in a dominant-negative manner, whereby the mutant protein adversely affects the function of the wild-type protein. These studies show that OB1/OB2 mutations diminish the interaction between POT1 and telomeric ss DNA in vitro, but do not generally affect telomerase recruitment to the telomere. Although some of the mutants do decrease telomerase activity in vitro, most variants are accompanied by telomere elongation. Both cell lines and patient samples harboring *POT1* mutations tend to have evidence of genomic instability, manifesting as increased telomere fragility, DDR, chromosomal aberrations, and alternative non-homologous end joining-mediated chromosomal fusions. These findings suggest a connection between the presence of *POT1* mutations and the occurrence of telomere dysfunction and genomic instability [50,69,70,72,89,93,99,109,110].

Mutations are less common in the C-terminal region of POT1 [26,70,71,80]. Functional experiments indicate that *POT1* C-terminal mutations could impact POT1 stability and disrupt its interaction with TPP1. *POT1* C-terminal mutants are also associated with an increased DDR and inappropriate DNA repair, suggesting that the POT1 N- and C-termini are both important for the maintenance of genome stability [26,27]. 

In summary, these findings suggest that mutant *POT1* might contribute to tumorigenesis in a manner that is dependent on telomere biology. POT1 alterations may introduce telomere dysfunction, leading to genome instability and ultimately promoting oncogenesis. As an example, malfunction of POT1 and DNA damage signaling in cardiac angiosarcoma has been suggested to assist in the acquisition of somatic mutations in the Vascular Endothelial Growth Factor (VEGF) pathway, thus facilitating cancer development [111].

## 4. Dysregulation of POT1 Expression in Cancer

POT1 expression in cancer has not been as well studied as *POT1* gene alterations. Some studies suggest that POT1 expression is upregulated in cancer samples compared to healthy controls. For example, *POT1* mRNA expression has been found upregulated in colorectal cancer, renal cell cancer, hepatocellular carcinoma, multiple myeloma and mantle cell lymphoma [112,113,114,115,116,117]. In contrast, *POT1* mRNA has been found downregulated in breast cancer [118,119]. In multiple cancers, POT1 expression has paradoxically been positively correlated with telomere length and telomerase activity, indicating a connection between POT1 dysregulation and cancer telomere pathology [112,120,121,122].

Within certain cancer types, the literature presents conflicting results. In CLL, POT1 has been shown to be overexpressed in early-stage CLL samples compared to normal cells [123]. However, *POT1* mRNA levels have been found to decrease in some B-CLL cases [124]. In gastric cancer, *POT1* mRNA expression detected by quantitative reverse transcriptase polymerase chain reaction (qRT-PCR) was significantly upregulated in tumors when compared with para-cancer tissues [122]. However, *POT1* mRNA measured by qRT-PCR tended to be downregulated in stage I/II tumors while remaining upregulated in stage III/IV cancers [120]. In a further independent study, POT1 protein expression detected by Western blot and immunohistochemistry was reduced in both gastric cancer cell lines and gastric tumors when compared with control samples [121]. 

Given the limited literature, it is difficult to draw consistent conclusions regarding POT1 expression in cancer. It is possible that POT1 expression is highly variable across cancer types, and also in some cases within cancer types. Apparently conflicting results could arise due to variable tumor biology, cohorts comprising heterogeneous cancers of differing subtypes, stages and grades, or could be due to differences in measurement approaches and the choice of normal controls.

Intriguingly, the downregulation of POT1 expression may contribute to the oncogenic role of Epstein–Barr Virus (EBV) in Hodgkin’s lymphoma: an EBV protein, Latent Membrane Protein 1 (LMP1), has been reported to downregulate three shelterin proteins by transcriptional and translational mechanisms and to cause the aggregation of telomeres and multinucleation [125]. The downregulation of TRF2 appears to be the most important component of this effect, but the downregulation of POT1 and TRF1 also contributes [125]. 

## 5. Potential Mechanisms of the Contribution of POT1 Dysfunction to Oncogenesis

The known functions of POT1 suggest that there may be multiple ways in which *POT1* mutations can contribute to the development of cancer (Figure 3). Some types of cancer are positively associated with telomere length. A number of groups such as the Telomeres Mendelian Randomization Collaboration [126] have analyzed genomic variants associated with telomere length at various loci, including POT1 [127], and have reported causal relationships between longer telomeres and the risk of cancers, including the following: melanoma, glioma, non-small-cell lung, serous ovarian, bladder, testicular germ cell, renal and endometrial [126]. It is interesting to speculate that this may because long telomeres allow a larger number of cell divisions while pro-oncogenic mutations are accumulating in nascent tumor cells, and that mutations in *POT1* may also have the effect of increasing telomere length in telomerase-positive tissue stem cells through a decreased ability of the mutant POT1 protein to inhibit telomerase activity. Moreover, it seems possible that the propensity for mutations at a range of POT1 amino acid residues to result in telomerase-dependent telomere lengthening accounts for this protein being mutated in cancer more often than other shelterin proteins.

Telomere dysfunction due to the partial loss of POT1 function may potentially arise from the decreased unfolding of telomeric G-quadruplexes, decreased inhibition of ATR signaling, DDR and inappropriate end-joining events, and decreased control of the C-rich strand fill-in reaction. This would contribute to genomic instability, and would be predicted to be particularly relevant to oncogenesis in cells that do not normally have telomerase activity (which would be capable of healing some or all of the telomere damage), such as the progenitor cells for many types of sarcomas. Moreover, the data regarding the suppression of telomeric recombination events by mouse Pot1 suggest the hypothesis that some *POT1* mutations may also facilitate activation of the ALT TMM, which is common in sarcomas. 

## 6. Clinical Implications of POT1 Alterations in Cancer

A greater understanding of telomere biology is expected to contribute to improvements in cancer management. TMMs are regarded as promising targets for cancer diagnosis, prognosis and therapeutics, and abnormalities in telomere biology such as short telomere syndromes have been associated with an increased risk of cancer development (reviewed by [128,129]). Although much more needs to be learnt about the role of shelterin proteins in human cancer, some clinical implications of *POT1* mutations are beginning to emerge.

### 6.1. Cancer Predisposition

Based on the data presented here, the inclusion of *POT1* in gene panels used for investigating cancer predisposition appears to be justified for familial melanoma, angiosarcoma, cardiac sarcoma, and CLL. The identification of predisposing variants has implications for prevention, surveillance and early detection, and genetic counseling (reviewed by [130]). Given the involvement of somatic *POT1* mutations in a small proportion of such a wide variety of cancer types (Figure 2), it seems possible that further studies of cancer predisposition genes will result in *POT1* being included in gene panels for many more types of familial cancer.

### 6.2. Cancer Prognosis

Identifying the presence of a somatic *POT1* variant may yield prognostic information for some cancers. *POT1* variants tend to indicate unfavorable prognosis. For example, the *POT1* variant rs35439397 has been associated with poor survival in breast cancer [131]. In survivors of childhood cancer, the presence of the *POT1* rs58722976 variant has been linked with an increased risk of developing a subsequent thyroid cancer [132]. In CLL, a four-gene panel that includes *POT1*, *XPO1*, *MYC88* and *BIRC3* can predict reduced survival in CLL and monoclonal B-cell lymphocytosis [83]. Importantly, since *POT1* mutations are found at a greater prevalence in relapsed CLL patients [90], the identification of *POT1* mutations in a CLL subclone could guide treatment decisions. *POT1* variants have indeed been associated with clonal evolution, and might contribute to CLL progression [86,133]. Moreover, mutated *POT1* might serve as a prognostic factor for poor survival in CLL patients taking chlorambucil-based chemotherapy or chemoimmunotherapy [88]. 

POT1 expression levels might also be valuable for evaluating prognosis. For instance, in multiple myeloma, *POT1* mRNA levels have been associated with clinical stage and patient mortality [116]. In colorectal cancer, there is a significant association between POT1 expression and clinical features such as cancer stage, site of occurrence and lymph node metastasis [112]. Furthermore, in gastric cancer, POT1 expression has been associated with cancer stage [121]. Notwithstanding the inconsistent results in the current literature regarding POT1 expression in cancer, it is conceivable that it may become useful to include measurements of POT1 expression in the routine molecular profiling of cancer for prognostic purposes and decisions about treatment intensity or modality.

Some studies suggest that POT1 is overexpressed in radioresistant cell lines of diverse cancer types [134,135,136,137,138]. In these cells, POT1 knockdown results in decreased radioresistance [134,137]. It has been suggested that increased POT1 levels may identify glioblastomas that would benefit from carbon ion hadron therapy [139].

### 6.3. Cancer Therapeutics

Various strategies are being developed to target the TMMs, telomerase reviewed by [128] and ALT [140,141,142,143]. The apparently complex relationship between *POT1* mutations and oncogenesis makes POT1 a difficult treatment target at present. Drug screening to find potential inhibitors of POT1 led to the discovery of a small molecular inhibitor known as bis-azo dye Congo red, which can suppress the interaction between ss DNA and POT1 through competitive binding to the POT1 N region [144,145]. However, the therapeutic value of POT1 inhibition in cancer is currently unknown. 

## 7. Conclusions

Germline and somatic *POT1* mutations have been identified in a range of cancer types. *POT1* mutations are enriched in the N-terminal OB-fold domains, and may contribute to tumorigenesis in a telomere-dependent manner. Deleterious *POT1* mutations are present in approximately 3% of all tumors, with a higher prevalence in angiosarcomas, non-small-cell lung cancers, and cutaneous squamous cell carcinomas and melanomas. *POT1* germline variants are associated with cancer susceptibility in multiple familial cancer types, most notably in melanoma. We speculate here that the mechanisms whereby *POT1* mutations contribute may differ depending on whether the progenitor cells have telomerase activity, and that the predominant effects of partial loss of POT1 function are increased telomere length in cells with telomerase activity, together with genomic instability. Genotype–phenotype correlation studies, and functional analyses of the biological consequences of cancer-associated *POT1* mutations, would advance this area of research.

The presence of germline *POT1* mutations has implications for genetic counseling, surveillance and early detection, and it is possible that the presence of *POT1* mutations and/or dysregulated expression of POT1 may yield useful prognostic information. Given that most cancer-associated *POT1* mutations are likely to result in a loss of function, the inhibition of POT1 might not be a useful therapeutic strategy, but an analysis of the genetic dependencies of *POT1* mutant cancer cells may suggest an effective synthetic-lethal approach to treatment.

## Figures and Tables

**Figure 1 cancers-12-02739-f001:**
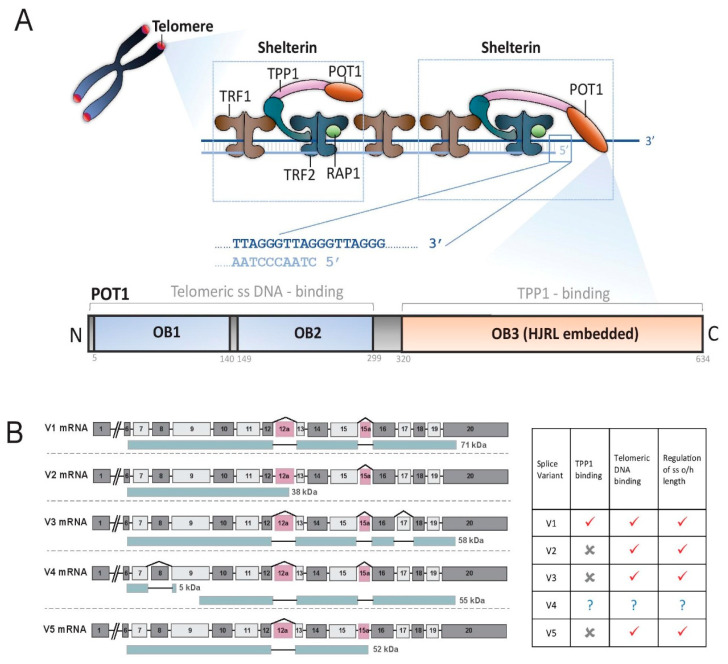
Shelterin complex and Protection of Telomeres 1 (POT1) protein. (**A**) Telomeres are composed of tandem repetitive TTAGGG sequences that are predominantly double-stranded (ds) but terminate in a single stranded (ss) 3′-overhang of the G-rich strand [6,36]. Telomeres are shielded with the shelterin complex, consisting of the TRF1, TRF2, RAP1, TIN2, TPP1 and POT1 proteins. POT1 binds to the TPP1 protein and to telomeric ss DNA through its C- and N-termini, respectively, and controls the sequence at the ds/ss junction. The human POT1 protein contains three oligonucleotide/oligosaccharide-binding (OB) fold domains, namely OB1-3, and OB3 has an embedded Holliday junction resolvase-like (HJRL) domain. (**B**) POT1 splice variants and their functions [21,29,37]

**Figure 2 cancers-12-02739-f002:**
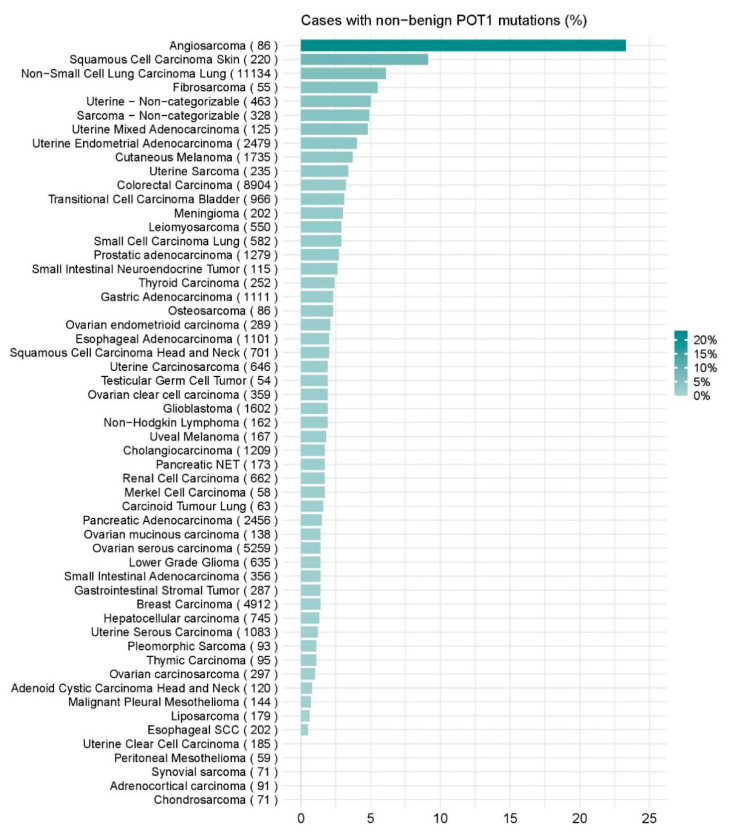
Burden of non-benign *POT1* mutations across cancer types. The prevalence of *POT* mutations defined as neither “presumed benign” nor “benign” was investigated in 62,368 tumors. The figure shows the *POT1* mutation burden in some tumor categories that included more than 50 cases (data from [65]). The number in brackets indicate the total number of cases for each cancer type.

**Figure 3 cancers-12-02739-f003:**
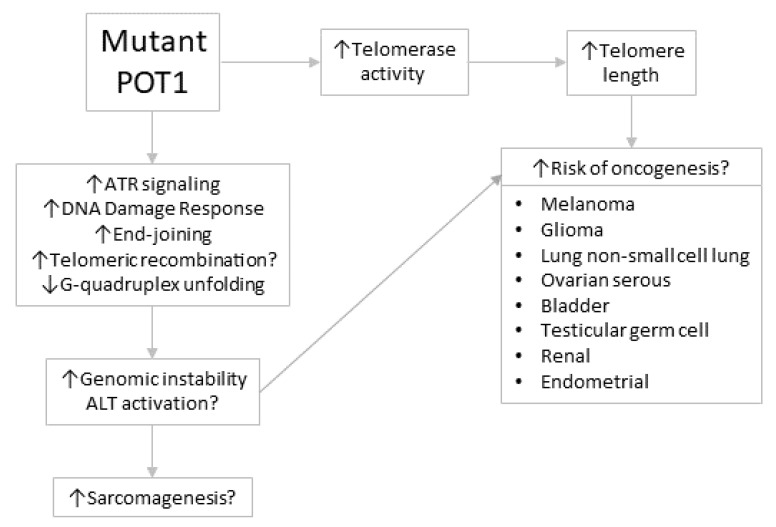
Proposed roles of *POT1* mutations in development of cancer. We speculate here that the effects of *POT1* mutations differ according to the telomerase status of the cancer progenitor cells, with genomic instability and perhaps the activation of alternative lengthening of telomeres (ALT) being the predominant contributor to oncogenesis in cells that are telomerase-negative and therefore less able to counteract the deleterious effects of the *POT1* mutation on telomere cap function, and with increased telomere length being the predominant contributor when the progenitor cells have telomerase activity.

**Table 1 cancers-12-02739-t001:** Prevalence of germline and sporadic POT1 mutations in selected cancer types.

Tumor Type	Sporadic Tumors		Hereditary Cancer Pedigrees	
	Prevalence	References	Prevalence	References
Cutaneous melanoma	3.8% (82/2179)	[65,76,77]	1.3% (21/1654)	[69,70,71,72,73,74,75]
Chronic lymphocytic leukemia (CLL)	2.5–10%	[81,82,83,84,85,86,87,88,89]	6.0% (4/66)	[80]
Angiosarcoma	19.8% (24/121)	[65,94,95]	20.0% (2/10)	[94]
Glioma	1.7% (39/2237)	[65]	5.3% (4/75)	[96,97]
Colorectal cancer	3.2% (281/8904)	[65]	0.3% (3/1194)	[98]
Osteosarcoma	2.3% (2/86)	[65]	0.5% (5/1004)	[102]

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
