# Peer review of "Role of POT1 in Human Cancer"

_cancers, 2020, doi:10.3390/cancers12102739_

Round 1
Reviewer 1 Report
Telomeres are specialized nucleoprotein structures located at the ends of linear chromosomes. They are crucial to guarantee the genome integrity as they protect chromosome ends from being recognized as DNA double-strand breaks (i.e., maintenance of genome stability) and in controlling the replicative potential of cells (i.e., limit the cell lifespan).
Two telomere maintenance mechanisms have been identified to date in human cancers, namely telomerase activity and the alternative lengthening of telomeres (ALT) pathway. The maintenance of telomere length and integrity has been described as an essential feature by which cancer cells attain replicative immortality (i.e., limitless lifespan) and stabilize their rearranged genomes.
The study of telomere biology is important for understanding cancer risk, diagnosis, prognosis and, possibly, the outcome of therapy.
In the present review papers, the authors focus on Protection of Telomere 1 (POT1), a component of the shelterin complex, a telomere-associated six-protein array that, together with ancillary factors, regulates telomere homeostasis by stabilizing telomere architecture and contributing to telomere protection and replication. In particular, the authors reviewed the biological functions of POT1, the prevalence of POT1 germline and somatic mutations as well as dysregulation of its expression across different cancer types.
Interestingly, the authors also propose a framework to explain how POT1 abnormalities may contribute to oncogenesis in different cell types as a function of the presence or of the absence of telomerase activity (Figure 3). Finally, the possible clinical implications of POT1 alterations in cancer as well as the possible approaches for the future development of targeted therapies have been also highlighted.
Overall, the manuscript is well-written, concise and pretty focused. Considering the meagerness of available literature data, this brand-new topic is well presented and adequately faced.
Minor concerns:
1) A table summarizing the prevalence of germline and somatic POT1 mutations in the different tumor settings (hereditary, sporadic and cancer predisposition syndromes) should be included at the end of section 3;
2) Similarly, a table summarizing the implications of POT1 alterations in cancer predisposition and prognosis should be included at the end of section 6;
3) In figure 1, a panel showing a graphical representation of the distinct characteristics of POT1 V1-V5 splicing variants should be included;
4) Line 189: the concept of “non-benign mutations” should be explained;
5) Check for typos.
Author Response
Replies are indicated in bold font.
Telomeres are specialized nucleoprotein structures located at the ends of linear chromosomes. They are crucial to guarantee the genome integrity as they protect chromosome ends from being recognized as DNA double-strand breaks (i.e., maintenance of genome stability) and in controlling the replicative potential of cells (i.e., limit the cell lifespan).
Two telomere maintenance mechanisms have been identified to date in human cancers, namely telomerase activity and the alternative lengthening of telomeres (ALT) pathway. The maintenance of telomere length and integrity has been described as an essential feature by which cancer cells attain replicative immortality (i.e., limitless lifespan) and stabilize their rearranged genomes.
The study of telomere biology is important for understanding cancer risk, diagnosis, prognosis and, possibly, the outcome of therapy.
In the present review papers, the authors focus on Protection of Telomere 1 (POT1), a component of the shelterin complex, a telomere-associated six-protein array that, together with ancillary factors, regulates telomere homeostasis by stabilizing telomere architecture and contributing to telomere protection and replication. In particular, the authors reviewed the biological functions of POT1, the prevalence of POT1 germline and somatic mutations as well as dysregulation of its expression across different cancer types.
Interestingly, the authors also propose a framework to explain how POT1 abnormalities may contribute to oncogenesis in different cell types as a function of the presence or of the absence of telomerase activity (Figure 3). Finally, the possible clinical implications of POT1 alterations in cancer as well as the possible approaches for the future development of targeted therapies have been also highlighted.
Overall, the manuscript is well-written, concise and pretty focused. Considering the meagerness of available literature data, this brand-new topic is well presented and adequately faced.
Minor concerns:
1) A table summarizing the prevalence of germline and somatic POT1 mutations in the different tumor settings (hereditary, sporadic and cancer predisposition syndromes) should be included at the end of section 3. We have added a Table (starting on line 297), that summarizes the data for a selection of cancer types where the prevalence of germline and sporadic mutations has been studied.
2) Similarly, a table summarizing the implications of POT1 alterations in cancer predisposition and prognosis should be included at the end of section 6. The data on these points are quite weak, and it is our view that including these data in a Table would give this section of the manuscript undue prominence.
3) In figure 1, a panel showing a graphical representation of the distinct characteristics of POT1 V1-V5 splicing variants should be included. Panel B has been added to Figure 1, which illustrates the splicing variants and summarizes their biological functions. The legend to Figure 1 (starting on line 103) has been modified accordingly, and the new panel has been referred to in the text (line 79).
4) Line 189: the concept of “non-benign mutations” should be explained. This concept has been defined in a sentence, starting on line 192, which has been added. Additional words relating to this concept have been added to the legend for Figure 2.
5) Check for typos.
Line 373 Q-quadruplex has been corrected to G-quadruplex; and a missing space has been inserted on line 420. No other typos have been identified.
Reviewer 2 Report
In this manuscript entitled "Role of POT1 in human cancer", the authors concisely summarized the function of POT1, a telomere ssDNA binding protein, in telomere maintenance and germline genetic mutations of POT1, as well as somatic POT1 mutations, identified across various cancer types. The authors then discuss the potential contribution of such POT1 mutations and dysfunctions to tumorigenesis in telomerase-positive and negative cancers. They also argue the clinical implication of POT1 mutations in terms of cancer predisposition, prognosis, and therapeutics. This review is very well written and comprehensively summarizes the publically available genetic mutations of POT1. I support this review's publication in the current form, while the following comments/suggestions may further improve the manuscript.
Minor points,
On page 8, line 292-, "most POT1 variants in cancer seem to arise at the N-terminal OB-fold domains". Because there are many genetic mutations of POT1 described in this review, it would be helpful to summarize the mutations as a diagram or table. For example, how about superimposing positions of the mutated residues on the POT1 protein diagram (as found in Figure 1)? If there are too many mutated residues, mutations from different cancer types can be shown as independent diagrams.
On page 1, line 13-, "POT1, the shelterin protein that appears to be most commonly involved in cancers". It would be nice if the authors provide a discussion on why POT1, among other shelterin components, is most commonly mutated in cancers.
In figure 3, what is the difference between a solid arrow and a dashed-arrow?
On page 10, line 357, Q-quadruplexes should be G-quadruplexes.
Author Response
Comments and Suggestions for Authors
In this manuscript entitled "Role of POT1 in human cancer", the authors concisely summarized the function of POT1, a telomere ssDNA binding protein, in telomere maintenance and germline genetic mutations of POT1, as well as somatic POT1 mutations, identified across various cancer types. The authors then discuss the potential contribution of such POT1 mutations and dysfunctions to tumorigenesis in telomerase-positive and negative cancers. They also argue the clinical implication of POT1 mutations in terms of cancer predisposition, prognosis, and therapeutics. This review is very well written and comprehensively summarizes the publically available genetic mutations of POT1. I support this review's publication in the current form, while the following comments/suggestions may further improve the manuscript.
Minor points,
On page 8, line 292-, "most POT1 variants in cancer seem to arise at the N-terminal OB-fold domains". Because there are many genetic mutations of POT1 described in this review, it would be helpful to summarize the mutations as a diagram or table. For example, how about superimposing positions of the mutated residues on the POT1 protein diagram (as found in Figure 1)? If there are too many mutated residues, mutations from different cancer types can be shown as independent diagrams.
We agree such a diagram would be very useful but, regrettably, many of the primary papers do not contain sufficient information to make this possible. We would need to approach the authors of a number of papers to obtain the data, which we expect could take several months to acquire.
On page 1, line 13-, "POT1, the shelterin protein that appears to be most commonly involved in cancers". It would be nice if the authors provide a discussion on why POT1, among other shelterin components, is most commonly mutated in cancers.
A discussion of this point has been added – sentence starting on line 371.
In figure 3, what is the difference between a solid arrow and a dashed-arrow?
The dashed arrow has been replaced with a solid arrow.
On page 10, line 357, Q-quadruplexes should be G-quadruplexes. This has been corrected (line 375 of revised manuscript).
Reviewer 3 Report
In this comprehensive review (based on 143 references) the authors describe and discus the current knowledge on POT1 - one of the shelterin proteins that appears to be most commonly involved in cancer. The manuscript is logically composed and authors sequentially describe interactions of the shelterin complex with telomeric DNA, describing in detail the functions of POT1, followed by germline and somatic POT1 mutations in cancer, including melanoma, CLL, angiosarcoma, glioma and other cancers. Special attention is paid to the dysregulation of POT1 expression in cancer and potential mechanisms of the contribution of POT1 dysfunction to oncogenesis as well as to clinical implication of POT1 alterations in cancer, including predisposition, prognosis and therapeutics. Paper is an interesting review not only updating and summarizing the current knowledge but also discussing new aspects in the understanding of a role of POT1 in human cancer.
In the manuscript authors are using a lot of abbreviations - majority with explanation, some without. As the review is orientated not only to the specialists in cell biology and molecular oncology, but could be interesting also to practical oncologists and other medical specialists, who are not so deeply into cell biology and molecular biology, it is desirable to decode the abbreviations as much as possible.
A few minor comments are following:
Line 17 - POT1 (Protection of Telomeres 1) - the explanation of the abbreviation is not identical to that on Line 70 - POT1 (Protection of Telomeres Protein 1) - must be identical;
Line 99 - ATR not explained (Ataxia telangiectasia- and Rad3- related);
Line 105 - RPA not explained (abundant heterotrimeric single-stranded DNA binding protein);
Line 107 - hnRNPA1 not explained (heterogenous nuclear ribonucleoprotein A1);
Line 108 - TERRA not explained (telomeric repeat-containing RNA);
Line 304 - The sentence needs to be corrected to understand it correctly, for example, "Fewer mutations localize in the C-terminal region of the POT1" or "Fewer mutations localize in the POT1 C-terminal region".
Author Response
Responses are indicated in bold.
Comments and Suggestions for Authors
In this comprehensive review (based on 143 references) the authors describe and discus the current knowledge on POT1 - one of the shelterin proteins that appears to be most commonly involved in cancer. The manuscript is logically composed and authors sequentially describe interactions of the shelterin complex with telomeric DNA, describing in detail the functions of POT1, followed by germline and somatic POT1 mutations in cancer, including melanoma, CLL, angiosarcoma, glioma and other cancers. Special attention is paid to the dysregulation of POT1 expression in cancer and potential mechanisms of the contribution of POT1 dysfunction to oncogenesis as well as to clinical implication of POT1 alterations in cancer, including predisposition, prognosis and therapeutics. Paper is an interesting review not only updating and summarizing the current knowledge but also discussing new aspects in the understanding of a role of POT1 in human cancer.
In the manuscript authors are using a lot of abbreviations - majority with explanation, some without. As the review is orientated not only to the specialists in cell biology and molecular oncology, but could be interesting also to practical oncologists and other medical specialists, who are not so deeply into cell biology and molecular biology, it is desirable to decode the abbreviations as much as possible.
A few minor comments are following:
Line 17 - POT1 (Protection of Telomeres 1) - the explanation of the abbreviation is not identical to that on Line 70 - POT1 (Protection of Telomeres Protein 1) - must be identical. This has been corrected (line 17 of the revised manuscript).
Line 99 - ATR not explained (Ataxia telangiectasia- and Rad3- related). This has been inserted (line 91 of the revised manuscript).
Line 105 - RPA not explained (abundant heterotrimeric single-stranded DNA binding protein). This has been inserted (line 110 of revised manuscript).
Line 107 - hnRNPA1 not explained (heterogenous nuclear ribonucleoprotein A1). This has been inserted (line 109 of revised manuscript).
Line 108 - TERRA not explained (telomeric repeat-containing RNA); This has been inserted (line 111 of revised manuscript).
Line 304 - The sentence needs to be corrected to understand it correctly, for example, "Fewer mutations localize in the C-terminal region of the POT1" or "Fewer mutations localize in the POT1C-terminal region". This sentence now reads "Mutations are less common in the C-terminal region of POT1" (line 314 of revised manuscript).
Reviewer 4 Report
The interaction between telomere maintenance and neoplasia, is a field of great interest from both fundamental and clinical perspectives. In this excellently written and structured manuscript, the authors review the important functions of the shelterin component POT1 in telomere homeostasis and genome integrity. They underpin the contribution of POT1 germline and somatic mutations across cancer predisposition syndromes and tumor types, and they discuss the dysregulation of POT1 expression in various types of cancers. The authors highlight differential effects of POT1 mutations according to TMM proposing a model for the detrimental effects of POT1 mutations on telomere protective function and uncontrolled telomere elongation. Finally, they discuss the clinical implications of POT1 in cancer diagnostics and the development of personalized oncotherapies. The manuscript contains highly informative illustrations, covers thoroughly the recent literature, and should be accepted as it is.
Minor comments
Lines 87-88: Figure 1 legend: “…POT1 binds to the TPP1 protein and to telomeric ss DNA through its N- and C-termini, respectively…” better change to “…POT1 binds to the TPP1 protein and to telomeric ss DNA through its C - and N -termini, respectively…”
Line 132 “…mutants that lack DNA binding yet retain TPP1 binding and thus have a dominant-negative function…” better be “…mutants that lack DNA binding yet retain TPP1 binding and thus having a dominant-negative function…”
Line 142 The phrase “…telomeres become lengthened when polα is inhibited partially by aphidicolin…” is confusing do you mean “…telomeres become lengthened when polα is partially inhibited by aphidicolin…”?
Line 357 Typo error “…unfolding of telomeric G-quadruplexes…”
Line 413 “…POT1 mutations are enriched in the N-termini OB-fold domains…” better be “…POT1 mutations are enriched in the N-terminal OB-fold domains…”
Author Response
Responses are indicated in bold font.
The interaction between telomere maintenance and neoplasia, is a field of great interest from both fundamental and clinical perspectives. In this excellently written and structured manuscript, the authors review the important functions of the shelterin component POT1 in telomere homeostasis and genome integrity. They underpin the contribution of POT1 germline and somatic mutations across cancer predisposition syndromes and tumor types, and they discuss the dysregulation of POT1 expression in various types of cancers. The authors highlight differential effects of POT1 mutations according to TMM proposing a model for the detrimental effects of POT1 mutations on telomere protective function and uncontrolled telomere elongation. Finally, they discuss the clinical implications of POT1 in cancer diagnostics and the development of personalized oncotherapies. The manuscript contains highly informative illustrations, covers thoroughly the recent literature, and should be accepted as it is.
Minor comments
Lines 87-88: Figure 1 legend: “…POT1 binds to the TPP1 protein and to telomeric ss DNA through its N- and C-termini, respectively…” better change to “…POT1 binds to the TPP1 protein and to telomeric ss DNA through its C - and N -termini, respectively…” This has been corrected (line 101 of revised manuscript).
Line 132 “…mutants that lack DNA binding yet retain TPP1 binding and thus have a dominant-negative function…” better be “…mutants that lack DNA binding yet retain TPP1 binding and thus having a dominant-negative function…” Changed to "mutants that lack DNA binding but retain TPP1 binding and thus have a dominant-negative function". (line 135 of revised manuscript).
Line 142 The phrase “…telomeres become lengthened when polα is inhibited partially by aphidicolin…” is confusing do you mean “…telomeres become lengthened when polα is partially inhibited by aphidicolin…”? Changed as recommended. (line 146 of revised manuscript).
Line 357 Typo error “…unfolding of telomeric G-quadruplexes…” Corrected (line 375 of revised manuscript)
Line 413 “…POT1 mutations are enriched in the N-termini OB-fold domains…” better be “…POT1 mutations are enriched in the N-terminal OB-fold domains…” Corrected (line 432 of revised manuscript).